# Relationship between Asian monsoon strength and transport of surface aerosols to the Asian Tropopause Aerosol Layer (ATAL): Interannual variability and decadal changes

Cheng Yuan[1,2], William K. M. Lau[2,3], Zhanqing Li[2,3], Maureen Cribb[2], Tijian Wang[1]

[1] School of Atmospheric Sciences, Nanjing University, Nanjing, 210023, China
[2] Earth System Science Interdisciplinary Center, University of Maryland, College Park, MD, 20740, USA
[3] Department of Atmospheric and Oceanic Sciences, University of Maryland, College Park, MD, 20740, USA

*Correspondence to*: Zhanqing Li (zhanqingli@msn.com) and William K.M. Lau (wkmlau@umd.edu)

**Abstract.** In this study, we have investigated the interannual variability and the decadal trend of carbon monoxide (CO), carbonaceous aerosols (CA), and mineral dust in the Asian Tropopause Aerosol Layer (ATAL) in relation to varying strengths of the South Asian summer monsoon (SASM) using MERRA2 reanalysis data (2001-2015). Results show that during this period, the aforementioned ATAL constituents exhibit strong interannual variability and rising trends connected to the variations of the strength of SASM. During strong monsoon years, the Asian Monsoon Anticyclone (AMA) is more expansive and shifted northward compared to weak years. In spite of effect of quenching of biomass burning emissions of CO and CA by increased precipitation, as well as the removal of CA and dust by increased washout from the surface to mid-troposphere in monsoon regions, all three constituents are found to be more abundant in an elongated accumulation zone at ATAL, on the southern flank of the expanded AMA. Enhanced transport to the ATAL by overshooting deep convection is found over preferred pathways along the foothills of the Himalayan-Gangetic Plain (HGP), and the Sichuan Basin (SB). The long-term positive trends of ATAL CO and CA are robust, while ATAL dust trend is weak due to its large interannual variability. The ATAL trends are associated with increasing strength of the AMA, with earlier and enhanced vertical transport of ATAL constituents by enhanced overshooting convection over the HGP and SB regions, out-weighing the strong reduction of CA and dust from surface to the mid-troposphere.

## 1 Introduction

The discovery from satellite lidar observations of the Asian Tropopause Aerosol Layer (ATAL) - a planetary-scale aerosol layer situated 13-18 km above sea level, spanning vast regions from the Middle East, South and East Asia to the western Pacific during the Asian Summer Monsoon (ASM) - has spurred active research on the composition ($H_2O$, chemical gaseous and aerosol species) and the relationship between the ATAL and the Asian Monsoon Anticyclone (AMA), and climate change (Fadnavis et al., 2013; Lelieveld et al., 2018; Li et al., 2005; Randel & Park (2006); Randel et al. (2010); Thomason and Vernier, 2013; Vernier et al., 2011; Vernier et al., 2015; Vernier et al., 2017; Yu et al., 2015). Previous studies have shown that deep convections in the tropics and volcanic eruptions can transport water vapor and surface pollutants including carbon monoxide (CO), sulfur dioxide ($SO_2$), and carbonaceous aerosols (CA) over source regions such as northern India and southwest China into the upper troposphere and lower stratosphere (UTLS) (Kremser et al., 2016; Li et al., 2005; Neely et al., 2014; Vogel et al., 2015). Other studies also reported that the ASM system can act as a conduit for these chemicals and aerosols convectively transported to the UTLS region (Bergman et al., 2013; Bergman et al., 2015; Bourassa et al., 2012; Garny and Randel, 2016).

Recent results from lidar observations, high-altitude balloon sounding data, and model simulations have shown a relatively higher concentration of chemicals and aerosols in the UTLS during the boreal summer, indicating effective vertical transport by the ASM (Babu et al., 2011; Kulkarni et al., 2008; Tobo et al., 2007; Yu et al., 2017). It has been suggested that, lifting tropospheric air parcels into the UTLS is associated with the establishment of the AMA during the peak phase (July-August) of the ASM (Gettelman et al., 2004; Park et al., 2007; Park et al., 2009; Ploeger et al., 2015; Randel and Park, 2006;

Randel et al., 2010). Upon entering the UTLS, gaseous chemical species and aerosols are advected anti-cyclonically and confined within the influence region of the AMA, forming the ATAL (Lau et al., 2018). Pan et al. (2016) found that the CO can be lifted into UTLS by deep convection over the southern flank of the Tibetan Plateau (TP) during boreal summer, and suggested that the dynamics of monsoon sub-seasonal variability may play important roles. Yu et al. (2015, 2017) found that

up to 15% of Northern Hemisphere UTLS aerosols came from vertical transport over the TP region via the ATAL during the ASM. Besides the Himalayan foothills, another transport pathway, located over central and southwestern China, has also been reported (Fadnavis et al., 2013). While UTLS transport processes have been shown to be closely related to the variability in the ASM, the mechanisms of UTLS transport processes and formation of the ATAL are not yet fully understood. A few recent studies have begun to examine relationship between ATAL and ASM on seasonal to sub-seasonal

time scales (Pan et al., 2016; Lau et al., 2018). However physical processes linking the ATAL and ASM on interannual and longer time scales are still unknown.

To recap, Lau et al. (2018), found a planetary scale "Double-Stem-Chimney-Cloud" (DSCC) encompassing two "stem regions": one over the Himalaya-Gangetic Plain (HGP) and the other over the Sichuan Basin (SB), where surface pollutants in Asian monsoon regions are pumped up to the UTLS during the boreal summer monsoon season, forming the

ATAL via turbulent mixing and advection by the large-scale anticyclonic circulation of the AMA. While heavy monsoon rain strongly removes aerosols by washout in the lower troposphere and near the surface, lofting by penetrative convection, anchored and amplified by orographic uplifting in the stem regions, can efficiently transport ambient aerosols in the middle and upper troposphere to the UTLS. They also found that the origin and variability of ATAL constituents, specifically CO, CA and dust, are closely linked to the seasonal development and intrinsic intraseasonal (20–30 days) oscillations of the

DSCC. This is a follow-up study to gain further new insights into physical processes leading to the ATAL variability on interannual to decadal time scales.

## 2 Data and analysis methods

### 2.1 Methods

Our study uses daily data from NASA's Modern Era Retrospective analysis for Research and Applications, Version

2 (MERRA-2) (Gelaro et al., 2017). This dataset is generated using the latest version of the Goddard Earth Observing System Model, Version 5 (GEOS-5) global data assimilation system, including the assimilation of aerosol optical depth (AOD) from MODerate resolution Imaging Spectroradiometer (MODIS) and Multi-angle Imaging Spectro-Radiometer (MISR) satellite retrievals. The MERRA-2 resolution is 0.5°x0.625° latitude-longitude with 72 vertical levels (Molod et al., 2015). It provides three-hourly global conventional meteorological data, i.e., temperature, winds, moisture, and precipitation,

as well as the concentrations of chemical gases and various aerosol species. All the processes of aerosol transport, deposition, microphysics, and radiative forcing are included. MERRA2 provides observations-based precipitation data, the product of precipitation has been assimilated and validated by both TRMM and GPCP (Reichle et al., 2017). Aerosol emissions from biomass burning and wildfires are derived from satellite Quick Fire Emission Dataset (QFED, (Darmenov and da Silva, 2013)). Anthropogenic aerosol emission inventory is from annual historical AeroCom Phase II (Diehl et al., 2012), up to the

mid-2000's depending on the availability of emission data for various gases and aerosol species (Randles et al., 2017). Beyond that the anthropogenic aerosol emissions are not updated. As such, the direct effects due to change in anthropogenic source emission cannot be assessed in MERRA2. The implication of this on our results will be discussed in the Conclusion in Section 4.

In this study, we choose CO, carbonaceous aerosols (CA) that include BC and OC, and dust as tracers for

diagnosing transport. Abundant quantities of CA and dust, found during the boreal summer season in the ASM region from local emissions and remote transport, could have strong impacts on the evolution of the Asian monsoon (Lau and Kim, 2006;

Lau et al., 2006; Lau, 2014; Meehl et al., 2008; Park et al., 2009; Vinoj et al., 2014). CO is a representative pollution tracer commonly used in previous studies of UTLS transport (Pan et al., 2016; Santee et al., 2017). This chemical gas is mainly emitted from biomass burning and industrial pollution Black carbon (BC) is a part of CA and is one of the main byproducts emitted from anthropogenic sources, as well as from natural wildfire activities. Organic carbon (OC), also a part of CA derived mostly from biomass burning and wildfires, is more abundant than BC in ASM regions (Chin et al., 2002), and has been detected at the ATAL (Yu et al., 2015). CA aerosols are not evenly distributed in the atmosphere like CO and is subject to wet and dry deposition. Emission sources of CO and CA, such as from local biomass burning can also be quenched by heavy monsoon rain (Lau, 2016; Lau et al., 2018). On the other hand, dust aerosols in ASM come from desert regions via long-range transport rather than from local emissions (Lau et al., 2008; Lau, 2014). This horizontal transport depends on the development of monsoon westerlies which extend from near the surface to the mid-troposphere (Gautam et al., 2009b; Lau et al., 2006; Zhang et al., 1996). While monsoon rain washout during the peak monsoon season (July-August) removes much of the coarse dust particles in and below clouds, ambient fine dust particles (< 0.2 μm) in and above clouds are lifted into the ATAL by penetrative deep convections anchored to the stem regions of the DSCC (Lau et al., 2018).

## 2.2 Data Availability

MERRA2 reanalysis data are available at https://disc.sci.gsfc.nasa.gov/daac-bin/ FTPSubset2.pl. The datasets processed and/or analysed during this study are available from the corresponding author upon reasonable request.

## 3 Results

### 3.1 Strong vs. Weak Monsoon

Figure 1a shows the climatological precipitation distribution and establishment of the AMA over the greater ASM region during the boreal summer monsoon season from July to August. The pronounced AMA with strong anticyclonic circulation (tropical easterlies and extratropical westerlies) develops in conjunction with heavy rainfall over the western Ghats of India, the Indo-Gangetic Plain (IGP) of North India, the Bay of Bengall, eastern China and the southeastern Asia region (Figure 1a). Additionally, the interannual variability of aerosols can be strongly affected by precipitation over the IGP region (Gautam et al., 2009a; Kim et al., 2016; Sanap and Pandithurai, 2015). In this study, we choose the domain (5°N–30°N, 70°N–95°E) to define strong vs. weak South Asian Summer Monsoon (SASM) years. This region is known to be subject to heavy monsoon precipitation, and orographic forcing which facilitates uplifting of water vapor, and atmospheric constituents by penetrative deep convection to the UTLS regions and above (Houze et al., 2007; Medina et al., 2010; Pan et al., 2016). The annual mean precipitation intensity for each year from 2001–2015 over the selected domain during the peak monsoon season (July-August) was calculated and used to represent the monsoon strength (Lau et al., 2000). Strong interannal variability and a robust increasing trend can be seen during this data period (Figure 1b). A simialr increasing decadal trend of the SASM has been reported in previous studies (Jin and Wang, 2017). To focus on interannual variability, we first detrended the rainfall time series, and then defined strong vs. weak monsoon years based on the detrended time series (Figure 1c). Strong (weak) monsoon years were selected when the mean rainfall was above (below) one standard deviation. Based on this procedure, four strong monsoon years (2007, 2010, 2011, and 2013; denoted as "SM") and three weak monsoon years (2002, 2014, and 2015; denoted as "WM") were identified. Composite mean distributions of monsoon meteorology, as well as aerosol loading transport, and ATAL variability were carried out for SM and WM respectively based on the detrended data. Henceforth, the term "anomaly" in the following parts refers to the difference between SM and WM composites (SM minus WM).

During SM, the AMA is stronger and more expansive than in WM, as evident in the corresponding 100-hPa geopotential height fields over the region (Figure 2a). The AMA in SM is wavier over the extra-tropics and appears to have shifted poleward, indicating a stronger extra-tropical influence on the AMA compared to WM years. The enhanced AMA in

SM occurs in conjunction with anomalous warming atmosphere above the TP and cooling in the lower stratosphere, as well as stronger anticyclonic circulation with anomalous westerlies at 35°N and easterlies at 20°N between 250–100 hPa, together with an elevated tropopause (Figure 2b). Cooling found near the surface is due to increased precipitation and cloudiness during SM. These are well known-features of a strong SASM monsoon (Huang and Sun, 1992; Lau et al., 2018; Randel and Park, 2006; Rodwell and Hoskins, 1996; Wang, 2006; Wu et al., 2007).

Figure 3 shows spatial distributions of climatological and anomalous rainfall, AOD, and low-level winds during July-August. Climatologically (Figure 3a), heavy rain (>6 mm day$^{-1}$) is found over the western Ghats, Bay of Bengal and Southeast Asia region. AOD is high over North Africa, the Middle East, and the Arabian Sea due to dust emissions from deserts, and transport by the southwesterly monsoon flow to the Indian subcontinent During SM years, enhanced precipitation is seen over the ASM land and adjoining oceanic regions of the Arabia Sea, and Indo-Western Pacific. The most pronounced increase is found over thewestern Ghatsof India and the HGP. Over East Asia, the presence of an elongated and southwest-northeast oriented dipole-like precipitation anomaly, together with the increased anticyclonic low-level circulation is indicative of a northward migration of the *Mei-yu* rain belt, associated with a strengthening of the subtropical high (Tao et al., 2001; Lau et al., 2000) (Figure 3b). Stronger low-level anomalous westerlies and easterlies are found over the Arabian Sea and the equatorial western Pacific, respectively. During SM, AOD is overall lower over Indian subcontinent and the tropical western Pacific due to stronger precipitation washout. Positive anomalous AOD are found over the Middle East and Central Asia. The former is related to increase surface emission of dust, and the latter is likely due to increased biomass burning emissions (Figure S1). Over East Asian, increase in AOD is found, possibly due to increased CA from biomass burning (Figure 3b, S1). Note that higher AOD and enhanced precipitation appear to coexist over northeastern China. This may be due to the aerosol swelling effect which is related to relatively higher relative humidity induced by enhanced *Mei-yu* rain belt during the moist summer monsoon season (Qu et al., 2016). Another possibility is that increased remote transport and uplifting above clouds by deep convection increased CA loading in the mid- to upper troposphere, even as CA in lower layers are removed by strong precipitation washout (Lau et al., 2018).

During SM, the 100 hPa geopotential height shows higher pressure over the subtropics and mid-latitude regions (25°N-40°N) with centers over the eastern (East Asia) and western ends (North Africa) portions of the climatological AMA (see Figure 2a). These high-pressure centers appear to be associated with a Rossby wave train pattern spanning the extra-tropics and the subtropics across Eurasia (Lau and Kim, 2012; Wang et al., 2008). Increased CO loading can be seen over three regions, *i.e.* North Africa, the TP, and central-northeastern China in an elongated "accumulation zone" along the southern flank of the expanded AMA (Figure 4b). For CA, similar centers of action can be found, except that regions of enhanced CA loading are more expansive and cover large parts of the AMA. Stronger concentration of CA is also seen along the southern flank of the expanded AMA, consistent with stronger easterly wind transport during SM years (Figure 4d, Figure 2b) (Lau et al., 2018). Higher loading of CO and CA can attribute to not only the deformation of the AMA, but also the enhancement of surface emission during SM years. As shown in the next subsection, during SM years, higher loadings of both CO and CA in the UTLS are found near regions of enhanced emissions only when there is increased vertical motion from deep convection (Figure S1). Similar to CA, more dust is also evident over the "accumulation zone" spanning North Africa, the Middle East, the TP, and East Asia during SM (Figure 4f).

## 3.2 Zonal and meridional cross-sections

In this subsection, we examine the changes in the ATAL structure along the axis of the DSCC (25°N-35°N) during SM and WM years. We begin with the structural changes in the vertical motion field under the influence of the AMA (Figure 5d). During SM years, overall enhanced anomalous ascending motions are found over the western sector (east of 85°E), while anomalous descending motions are found over the eastern sector (west of 85°E) of the AMA. In the western sector, two regions with strong vertical motion are found clustered over North Africa/the Middle East (15°E–50°E) and over

the foothills of the HGP (70°E–85°E) with anomalous ascent extending above 100 hPa in both regions. Over the western sector and embedded within a large region of overall anomalous descent, enhanced ascent is also found over East Asia around 105°E–115°E reaching above 100 hPa. As noted earlier (see Figure 3c), during SM years, the *Mei-yu* rain belt is shifted northward, leaving behind mostly anomalous descending motions in this latitudinal zone. However, moderately increased ascent is found over western central China (105°E–120°E) from the eastern foothills of the TP and the SB, collocating with the southern tip of the northward-shifted *Mei-Yu* rain belt. These three regions of anomalous ascent play essential roles in the distribution of chemical gases and aerosols species in the ATAL.

During SM years, the CO concentration is generally increased in the ATAL (Figure 5a), consistent with the enhanced advection by the strengthened easterlies at the southern flank of the AMA. Three centers of anomalous high CO concentration in the UTLS (200–100 hPa) over North Africa, the TP, and East Asia (identified in Figure 4b) stand out. These centers appear to be connected via stems of high CO related to the aforementioned three regions of anomalous ascent. The large reduction in CO near the surface over East Asia may be related to quenching of emission sources by increased precipitation over this region (Figure 5a, S1b). For CA, the pattern of anomalies is similar to the pattern of CO, with overall increased loading in the UTLS, and three action centers connected by stems of high CA to the surface (Figure 5b). The increase in near-surface CA over desert regions (east of 70°E) is consistent with increased surface emissions (Figure S1d). The reduction in CA in the monsoon region (west of 70°E) is likely due to stronger precipitation washout during SM. Likewise, during SM, severely suppressed dust is found near the surface up to the mid-troposphere in the stem over the HGP (60°E–100°E), associated with washout by the increased precipitation (Figure 5c, Figure 3b). Similar to CO and CA, dust reduction can also be seen in the middle and lower troposphere over east China (105°E–135°E), because of the enhanced rainout process. Due to the increased near-surface wind, dust loading is increased over the Middle East (30°E–70°E) but decreased over North Africa. Sources of dust contributing to the increased dust loading in the UTLS (above 200 hPa) seem mainly come from the Middle East/West Asia, with some contribution from the eastern TP, abutting the SB region.

Two meridional cross-sections (80°E-85°E and 100°E-105°E), respectively for the HGP (Figure 6) and the SB (Figure S2) regions have been examined. Because of similarity in patterns, only the HGP regions (Figure 6) are discussed here. Ascending motions during SM years over the HGP region near the foothills and top of the TP are enhanced and weakened locally in the vicinity of 20°N, associated with the enhancement and northward shifting of the AMA (Figure 6d). Additional increased ascending motions are south of 20°N, likely with the increased precipitation over the southern India and the northern Indian Ocean (see Figure 3b). A dipole pattern featuring increased CO over the top of the TP from 500 to 70 hPa at the northern edge of the climatological CO maxima coupled a reduced CO south of 20°N, Figure 6a again indicates that more CO was lifted into the UTLS by the enhanced vertical motion associate with the northward shift of the AMA during SM years. The reduction of CO in the lower troposphere and near the surface in the extratropics (40°N-58°N) is likely related to the quenching of emission sources of biomass burning over the region (Figure S1b). Similar to CO, more CA is transported and enters the UTLS via the HGP stem in SM years, and the increased loading is more expansive than CO spanning 25°N-60°N, from 500 hPa to 50 hPa. This may be due to increase in biomass burning emission sources over northern Central Asia (Figure 6b and Figure S1d). Associated with the northward shifting of the AMA, CA concentrations below 100 hPa over the tropical region are substantially reduced. During SM years, dust is mostly reduced over the regions from surface to the upper troposphere. Increase uplifting of dusts into the UTLS by anomalous ascending motions are found over the TP and the Taklamakan desert (35°N-42°N). The pronounced reduction in CA and dust loadings over the foothills of the TP and the India subcontinent is due to wet scavenging effect by the enhanced rainfall over the region. For the SB stem region (Figure S2), the pattern of anomalous concentrations of CO, CA, and dust at the ATAL are similar to the HGP region, reflecting the competing influences of lofting by deep convection, emission quenching (for CO), and removal by precipitation washout (for CA and dust).

### 3.3 Long-term trends

To depict in long-term change in ATAL we have computed time series of CO, CA and dust averaged between 200-100 hPa layer, and over a large domain (60°E-120°E, 25°N-35°N), approximately bounding the AMA. For comparison, a time series representing the strength of the AMA, defined as the difference in zonal winds between northern (30°N-40°N) and southern (10°N-20°N) flanks of the AMA, has also been constructed (Figure 7). Clearly, CO and CA in the ATAL possess significant increasing trends during 2001-2015, at a rate of +7.8% (p-value = 0.018) and +12.7% per decade (p-value = 0.025), respectively. Both the CO and CA trends are consistent with a significant (p-value = 0.06) trend of AMA strength at a rate of +6.7% per decade. Given that the AMA is an essential component of the SASM, this suggests that the trends of increased loading of ATAL CO and CA could be attributed to the strengthening of the SASM during 2001-2015. For dust, the positive trend is weak with a rate of 1.6% per decade, and not significant (p-value = 0.875) due to the large interannual variability. The weak ATAL dust trend may be due to the removal of large fraction of dust particles by wet scavenging in and below raining clouds, out-weighting the effects of lofting by deep convection (Chin et al., 2000; Lau et al., 2018). Additionally, the large interannual variability of ATAL dust transport is also likely a reflection of the influence of non-monsoon factors, such as extratropical westerlies that can strongly affect long-range dust transport at high elevations (Sun et al., 2001; Huang et al., 2007).

To better understand the physical processes underpinning the ATAL long-term trend signal, we have constructed the time-mean vertical profiles of ATAL constituents, vertical motions and rainfall along critical east-west cross-sections spanning the AMA, respectively for the early period (EP, 2001-2006) and later period (LP, 2010-2015). The long-term change is defined as the difference between the two periods (LP minus EP). Figure 8a-d shows east-west cross-sections of long-term changes in CO, CA, dust, and vertical motions respectively, covering the same ASM region as in Figure 5. During LP, enhanced ascending motions (relative to EP) that reach the ATAL are most pronounced over the Pakistan/Northeast India, and the HGP region (60°E-95°E) (Figure 8d). A cluster of ascending motions are also found over greater SB regions of East Asia (100°E-130°E), in connection with the northward migration of the *Mei-yu* rainbelt (See Figure 3b). A third region of enhanced ascent is found over North Africa (15°E-30°E). During LP, overall, the CO concentration increases from the surface to the UTLS, with pockets of reduced CO near the surface due to biomass emission quenching by precipitation (Figure 8a). Similarly, CA concentration at the UTLS is increased during LP (Figure 8b), and appears to be connected to surface sources of increased CA over the North Africa/Middle East, and West Asia region (Figure S2) via the increased ascending motions over the Pakistan/NE India and HGP region (60°E-90°E). Strong reduction in CA from the surface to mid-troposphere found over East Asia (100°E-130°E) is due to the removal by increased precipitation washout. Compared to CO and CA, the increase in ATAL dust is modest (Figure 8c), and appears to follow a transport pathway from surface to the UTLS similar to CA. The increase in surface dust over the Middle East/West Asia region (40°E-70°E) may be related to a robust recent decadal warming trend over the India subcontinent and the Middle East (Jin and Wang, 2017). A hotter desert surface is likely to favor a deeper planetary boundary layer, enhanced dry convection and uplifting of dust from the surface (Gamo, 1996; Cuesta et al., 2009). During LP, overall reduction in dust from surface to mid-troposphere over monsoon regions is due to removal by increased precipitation washout.

Next, we examine the competing influences of lofting by overshooting convection and precipitation washing out in the DSCC stem regions (25°N-35°N, 65°E-115°E), including both the HGP and SB domains. ATAL trend, by examining the mean daily variations of monsoon precipitation, and vertical profiles of CO, CA, and dust over the region, during EP and LP respectively. During LP, monsoon precipitation is enhanced compared to EP from June through August (Figure9d, h), consistent with the increased rainfall trend shown in Figure 1b. CO concentrations from the surface to 200 hPa in LP is higher than in EP during the pre-monsoon period in May-mid-June (Figure 9a, e), reflecting a hotter land surface and enhanced dry convection over the region before monsoon onset. The onset of the monsoon as characterized by an abrupt rise in CO (region shaded by light yellow in Figure 9a, e) to above 200 hPa reaching the ATAL, occurs earlier in LP (around

June 16), compared to EP (around July 1). Thereafter, CO remains higher in LP, through the end of the monsoon season, maintaining a longer residence time at the ATAL, via the cumulative effect (multi-year mean) of lofting by deep convection. From the surface to the lower troposphere, CO concentration declines faster in LP, due to the quenching of emission by heavier monsoon rain. Likewise, for CA, features such as the earlier onset, the increased ATAL concentration (above 200 hPa), and the longer residence time during LP are also pronounced (Figure 9b, f). The competing influences of convective

lofting and wet removal can be seen in the more episodic increased in ATAL loading in both EP and LP, more so in the latter. During LP, the more efficient lofting of CA into the ATAL from the mid-troposphere during early July, and late August coincide approximately with the time of maximum precipitation, when deeper and more overshooting convection tend to occur (Figure 9f). During May in LP, a strong increase in CA from the surface to 200 hPa is noted. This could be related to a warming trend of the land surface over northern India and the desert regions to the west (Jin and Wang, 2017). A warmer

and drier land surface before monsoon onset is likely favor increased biomass burning emissions (van der Werf et al., 2006) (Figure S3). In contrast to CO and CA, dust concentration at ATAL varies littlefrom EP years to LP, with a slight signal of increased convective lofting during mid-July to mid-August in LP. This is consistent with the weak positive, but statistically insignificant dust trend shown in Figure 7. A notable signal is the increase in dust loading from surface to 300 hPa during May in LP compared to EP and rapid decline due to removal by washout during the June-August. A similar analysis

separately for the HGP and SB region have also been carried out. Results show that while the both regions exhibit the similar characteristics features regarding convective lofting and washout, the signal over the HGP is more pronounced than that over the SB region (Figure S4 and S5). This may be because the *Mei-Yu* rainfall system affecting the SB regions possesses more transient and migratory features compared to the more land-locked convection over the HGP region (Ding and Chan, 2005; Lau and Weng, 2001).

**4 Summary**

In this study, we have investigated the roles of monsoon physical processes in the interannual variability and long-term change of ATAL gaseous and aerosol species, *i.e.*, CO, carbonaceous aerosol (CA), and dust using 15 years (2001-2015) of NASA MERRA-2 reanalysis data. A monsoon index based on areal mean rainfall over the South Asia Summer Monsoon (SASM) region shows strong interannual variability and a robust long-term trend. Composite analyses were carried out

comparing strong monsoon years (SM) vs. weak monsoon years (WM) based on the detrended data. Regression trend analyses, and composite were carried out with the full data. During SM, the Asian monsoon anticyclone (AMA) is expanded, and shifted poleward relative to weak monsoon years, in conjunction with enhanced heating over the upper troposphere above of the TP, cooling in the lower stratosphere, and a rise of the tropopause height, relative to WM. During SM, more ambientCO, CA, and dust enter the ATAL from preferred pathways over the foothills of Himalayas-Gangetic Plain (HGP)

and the Sichuan Basin (SB). Upon entering the ATAL, these constituents are advected by the anomalous AMA circulation, which appears to be a component of a planetary scale Rossby wave train connecting the tropics and extratropics. As a result, enhanced loading of CO, CA and dust are found in an elongated accumulation zone on the southern flank of the extended AMA. During SM, enhanced UTLS transport of CO and CA to the ATAL, can be attributed to lofting by deep convection over the HGP and SBstem regions. While CO and CA, from surface to the mid-troposphere in the stem regions are reduced

during the peak monsoon season due toenhanced wet scavenging, more ambient CO and CA in the middle and upper troposphere continued to be transported into the ATAL due to increased overshooting convection. While stronger low-level westerlies transport more dust to the Indian subcontinent during SM, stronger precipitation washout suppresses dust loading near the surface in both the HGP and the SB stem regions. Dust over the West Asia/Middle East and the subtropical area in northwestern China contribute mostly to the dust enhancement in the UTLS.

We found robust positive significant decadal trends in CO and CA, as well as a weak positive but insignificant trend in dust in the ATAL. Overall, these trends are associated with an earlier onset of stronger overshooting convection over the HGP and SB regions, transporting ambient CO, CA and dust into the ATAL in conjunction with a strengthening of the Asian summer monsoon during 2001-2015. The increase in ATAL constituents occurs, even though there is reduction in surface CO due to emission quenching, and strong reduction in CA and dust due to increased precipitation washout in Asian monsoon regions during this period.

It is worthy to notify that there are limitations in using the MERRA2 aerosol species concentrations for intenrannual variability and long-term trend analysis. The MERRA2 system adjusts the model simulation according to the total AOD retrieved from satellite measurements during assimilation, but there is no speciated aerosol information from satellite data to allow changes of aerosol composition which is simulated by the widely-used chemical model of GOGART (Chin, 2000, 2002, 2016; Kim, 2017). As a result, all model simulated aerosol species had to be adjusted by the same factor, which can introduce artifacts for increase or decrease of individual aerosol mass or AOD (Randles et al., 2017). To test if the interannual variability or long-term trends of individual aerosol species inferred from MERRA2 might be contaminated by any non-physical corrections of individual aerosol species during the assimilation process. We have taken a look at the 'increment' for CA and Dust from the MERRA2 dataset. Results show that in our research domain, the assimilation increments for CA and Dust aerosols are very small. In most cases, it is nearly zero and the ratio of the rest increment to the values of the model mean signal is less than 1%. Therefore, the model aerosol physics are likely to be reasonable.

As a caveat, we note that while we have found overall significant relationships connecting interannual variability and long-term trends in ATAL constituent transport processes and monsoon strength, this study leaves open the question of how changes in anthropogenic emissions may affect the relationships. This is because the MERRA2 emission inventories of aerosols species have not been updated after the mid-2000's (Randles et al., 2017). Moreover, recent modeling studies have suggested that the mixing state and aging processes can largely change the aerosol lifetime during simulation, and consequently affect the amount of aerosols lifted to UTLS, and some optical measurements further support that dust aerosol can be coated by anthropogenic aerosols over East Asia and then significantly enhance absorbing ability (Wang et al., 2018, Tian et al., 2018). Nonetheless, our findings provide a working hypothesis that warrants further investigations using both modeling and observational studies. Long-term "top-down" satellite observations, and "bottom-up" field observations including updated emission inventories, as well as intercomparison among climate models with state-of-the-art representation of aerosol physics and chemistry will be needed to test our hypothesis.

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

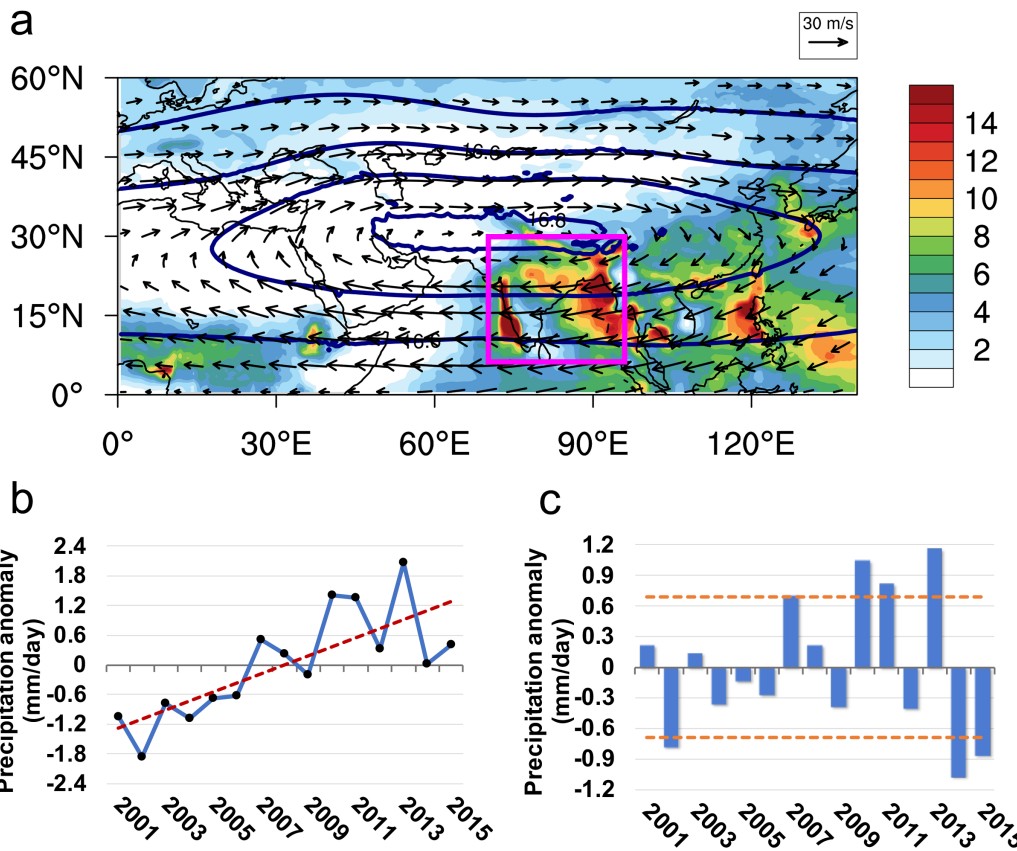

**Figure 1. Climatological mean ASM features associated with the AMA showing (a) the spatial distribution of winds (arrows, in m s⁻¹), geopotential height at 100 hPa (solid contours, in km), and rainfall (colored background, in mm day⁻¹) during July-August of 2001-2015. The pink box (5°N-30°N, 70°N-95°E) outlines the domain selected for calculating the precipitation intensity. (b) Time series of the precipitation anomaly from 2001-2015 (with the trend line in red). (c) The de-trended distribution (with standard deviations in orange).**

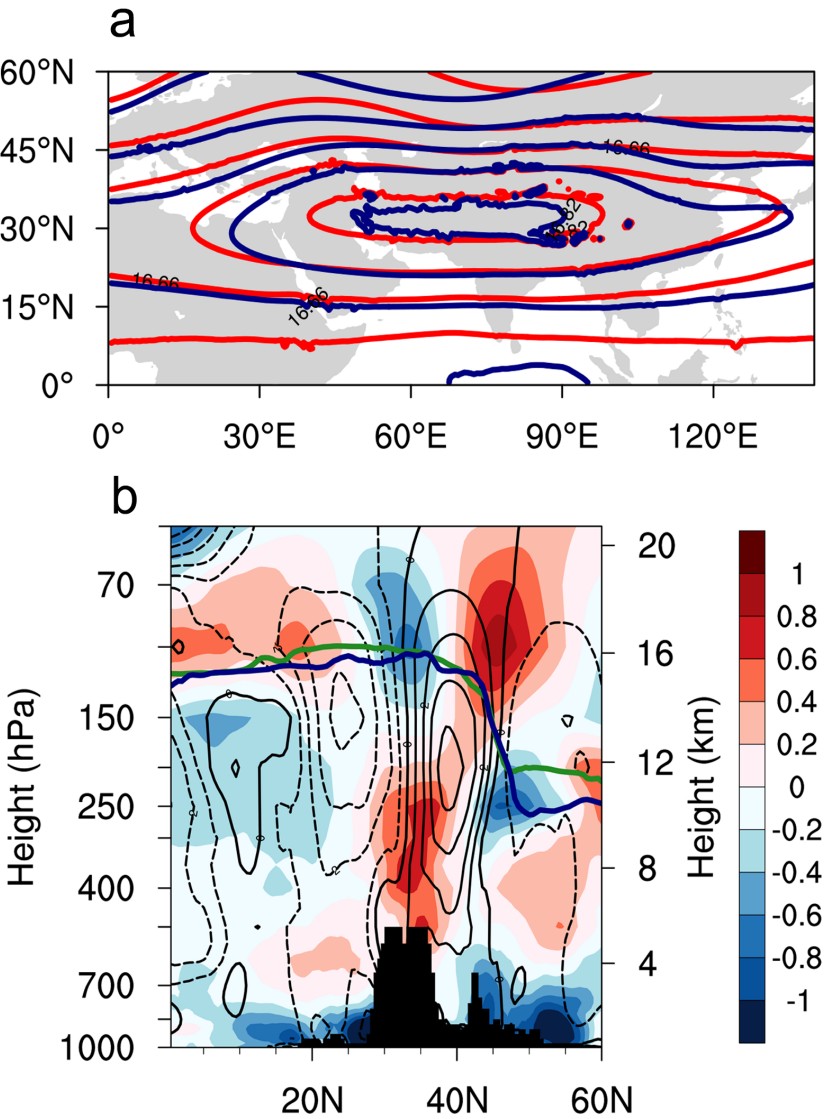

**Figure 2.** (a) 100-hPa Geopotential height (in km) in strong (weak) monsoon years during July-August is shown in red (blue). (b) Latitude-height cross-section of temperature anomalies (color shaded, in K), zonal wind anomalies (contour lines, in m s⁻¹) between strong and weak monsoon years ('strong' minus 'weak'), and tropopause height (thick lines) in strong monsoon years (green) and weak monsoon years (blue) over the Indian subcontinent (80°E-85°E) during July-August.




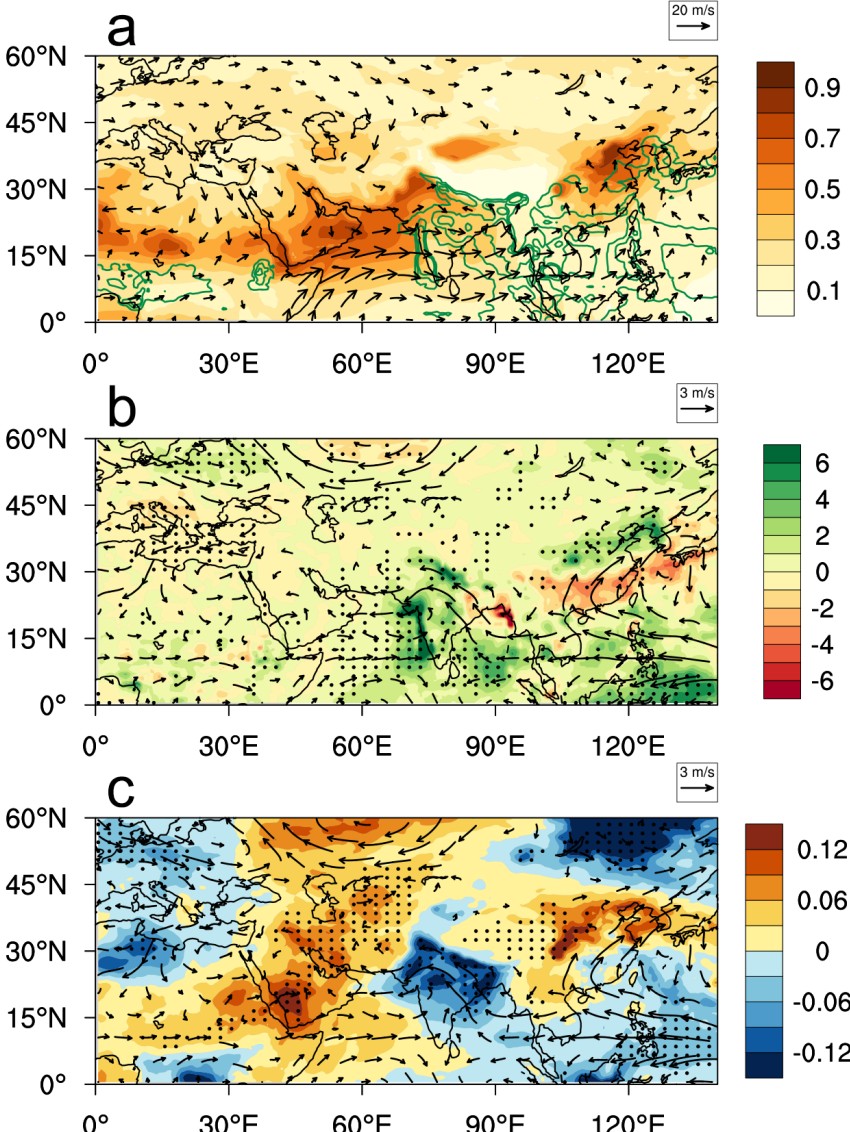

Figure 3. (a) Spatial distributions of climatological AOD during July to August, superimposed with precipitation (only those >6 mm day⁻¹ are shown) and 850 hPa wind (arrows, in m s⁻¹) (b) Spatial distributions of anomalous ('strong' minus 'weak') precipitation (mm day⁻¹) and 850 hPa wind (arrows, in m s⁻¹). (c) is same as (b) except the patter is showing anomalous AOD and 850 hPa wind (arrows, in m s⁻¹). Dots represent data points with a significance > 95%.


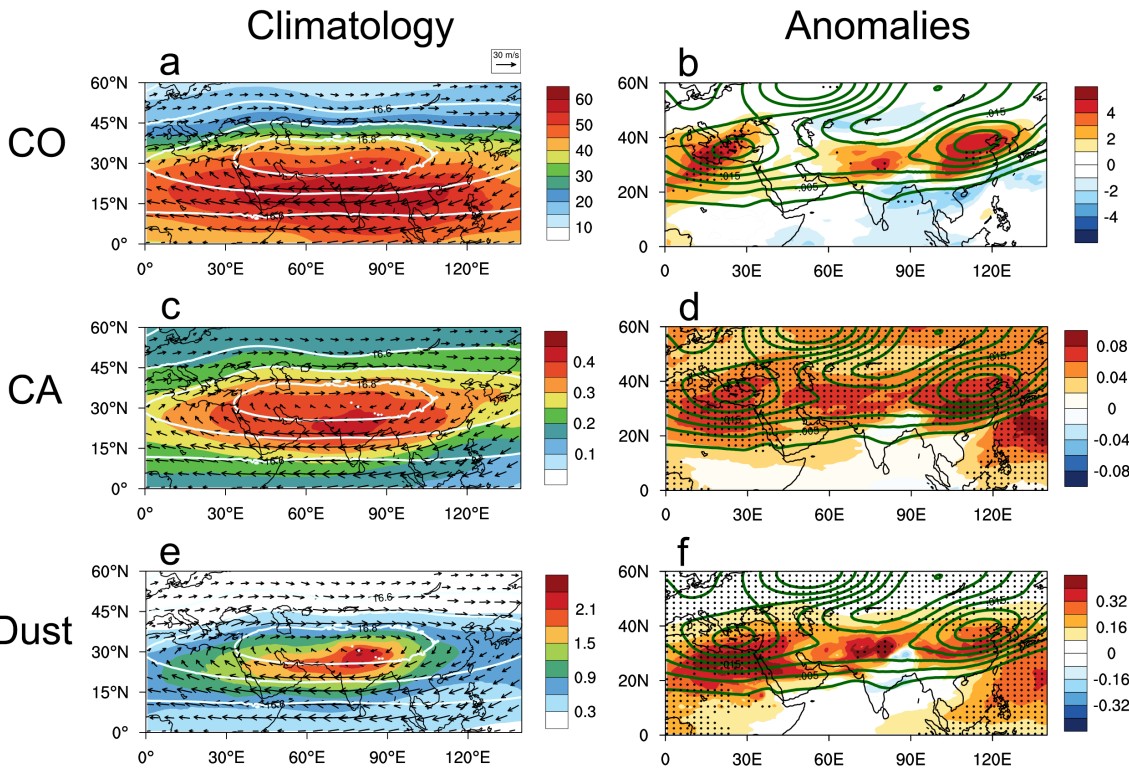

**Figure 4.** Spatial patterns of chemical gases and aerosols distributions of (a) CO (ppbv), (c) CA (ppbm), and (e) dust (ppbm) at 108.7 hPa during July-August, superimposed with geopotential height anomalies at 100-hPa (white contours, in km) and 108.7 hPa winds (arrows, in m s$^{-1}$). Panels (b), (d), and (f) are the same as (a), (c), and (e) except that they show anomalous distributions between strong and weak monsoon years ('strong' minus 'weak'), superimposed with geopotential height anomalies (green contours) at the same level. Dots represent data points with a significance >95%.

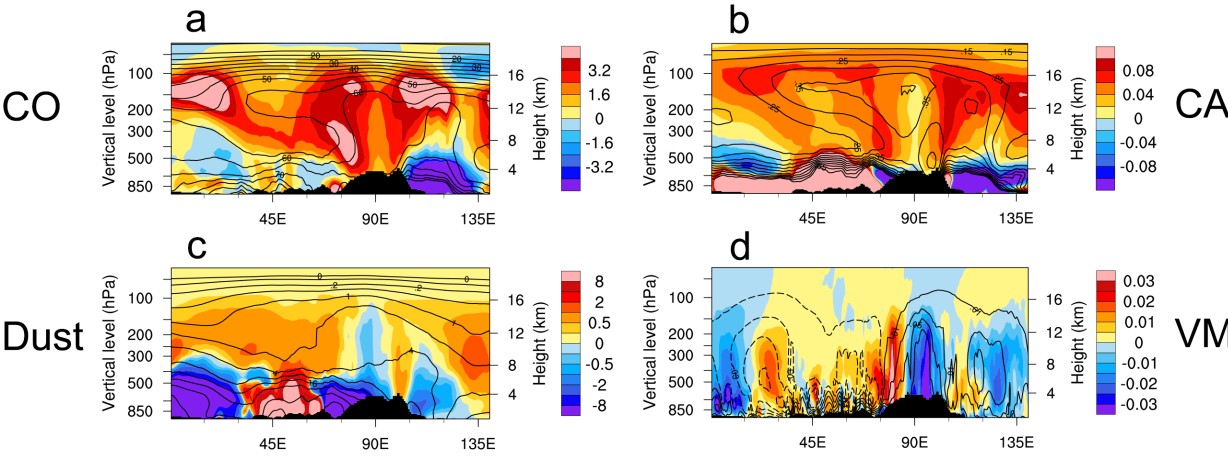

**Figure 5.** Longitude-height cross-sections (0°E-140°E) of (a) CO (ppbv), (b) CA (ppbm), (c) dust (ppbm), and (d) vertical motion (Pa s⁻¹) anomalies between strong and weak monsoon years ('strong' minus 'weak') averaged over the southern portion of the AMA (25°N-35°N) during July-August, superimposed with the climatological mean of weak monsoon years (black contours). For vertical motions in (d), solid (dashed) contours indicating ascent (descent).

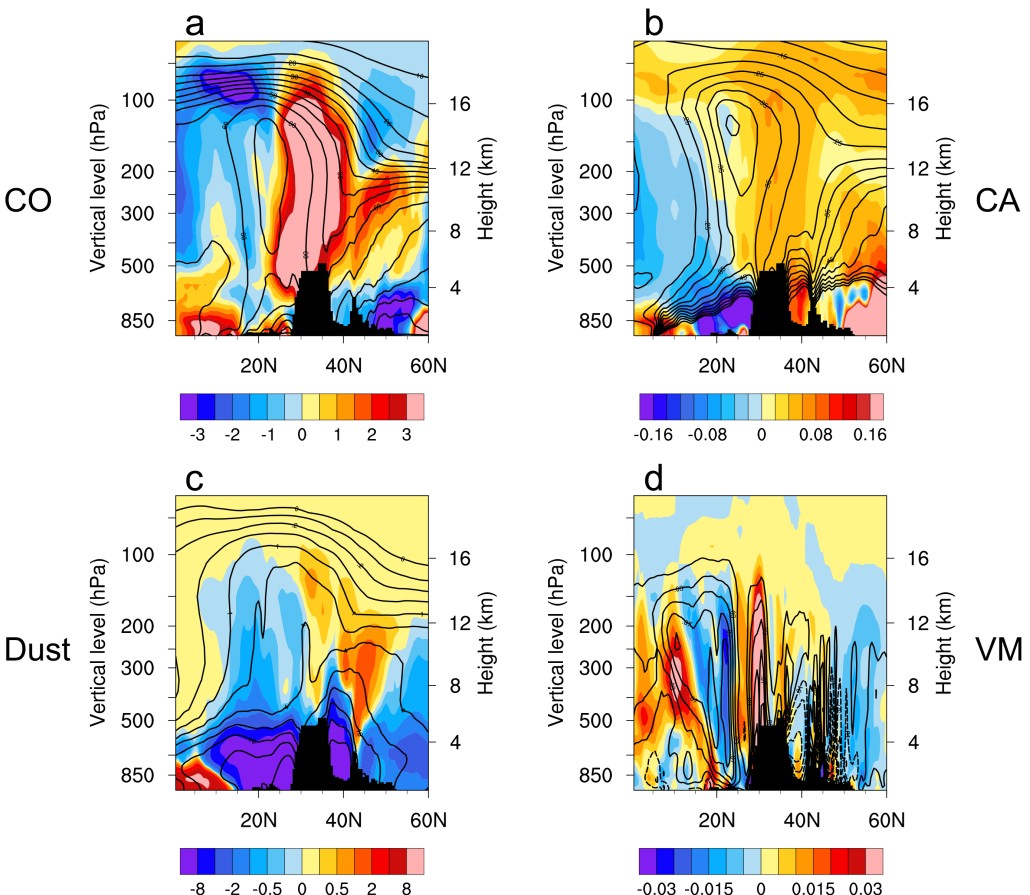

**Figure 6.** Latitude-height cross-sections (0°N-60°N) of (a) CO (ppbv), (b) CA (ppbm), (c) dust (ppbm), and (d) vertical motion (Pa s⁻¹) anomalies between strong and weak monsoon years ('strong' minus 'weak') averaged over the HGP region (80°E-85°E) during July-August, superimposed with the climatological mean of weak monsoon years (black contours). For vertical motions in (d), solid (dashed) contours indicating ascent (descent).

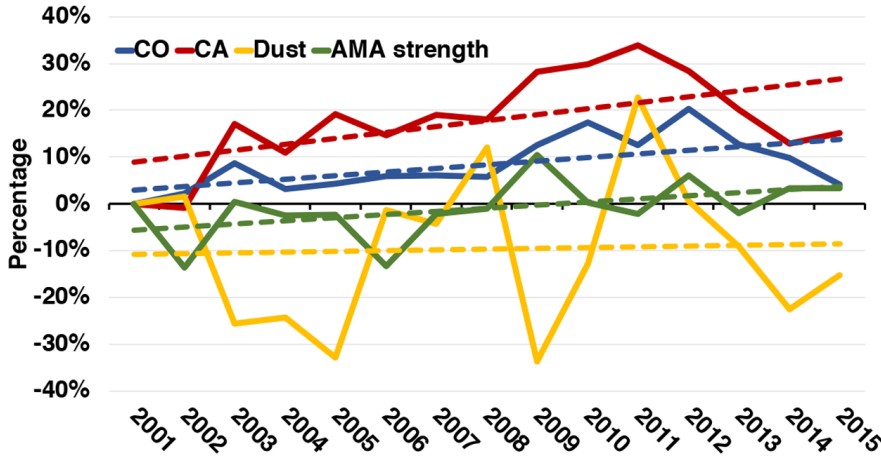

| | Increasing rate (percentage/per decade) | P value |
|---|---|---|
| AMA strength | 6.7 | 0.06 |
| CO | 7.8 | 0.018 |
| CA | 12.7 | 0.025 |
| Dust | 1.6 | 0.875 |


**Figure 7. Time series of CO, CA, dust and AMA strength anomalies (percentage) relative to the first year during 2001-2015. The loading of CO, CA and dust is area-averaged over selected region (that is, 25°N-35°N, 60°E-120°E). The AMA strength is calculated from the percentage difference in zonal wind averaged between 30°N-40°N minus zonal wind averaged between 10°N-20°N along the sector 60°-120°E. The increasing rate and p-value from significant**

**test for each variable are shown in the lower box.**


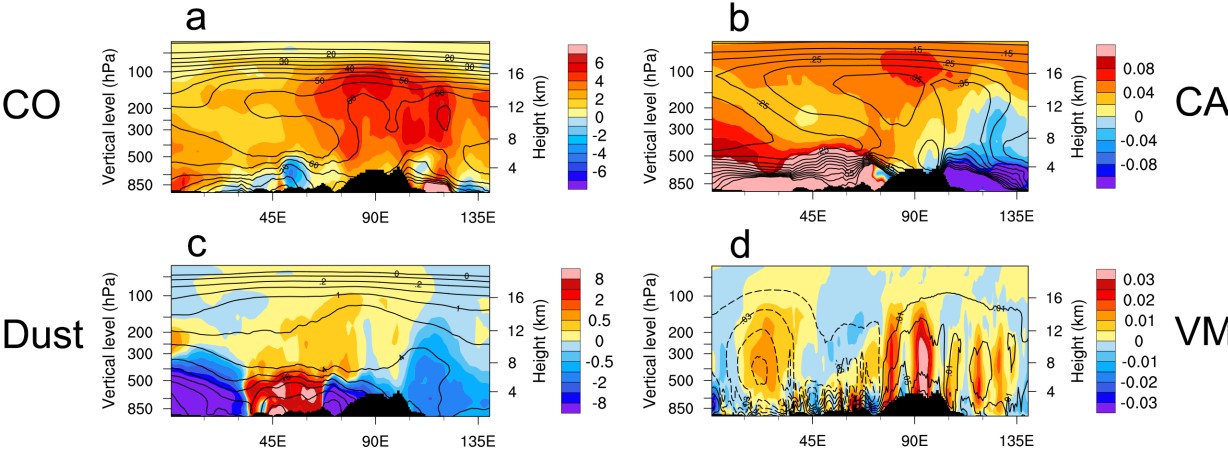

**Figure 8.** Longitude-height cross-sections (0°E-140°E) of (a) CO (ppbv), (b) CA (ppbm), (c) dust (ppbm), and (d) vertical motion (Pa s⁻¹) anomalies between Late Part years and Early Part years ('Late' minus 'Early') averaged over the southern portion of the AMA (25°N-35°N) during July-August, superimposed with the climatological mean of Early Part years (black contours). For vertical motions in (d), solid (dashed) contours indicating ascent (descent).






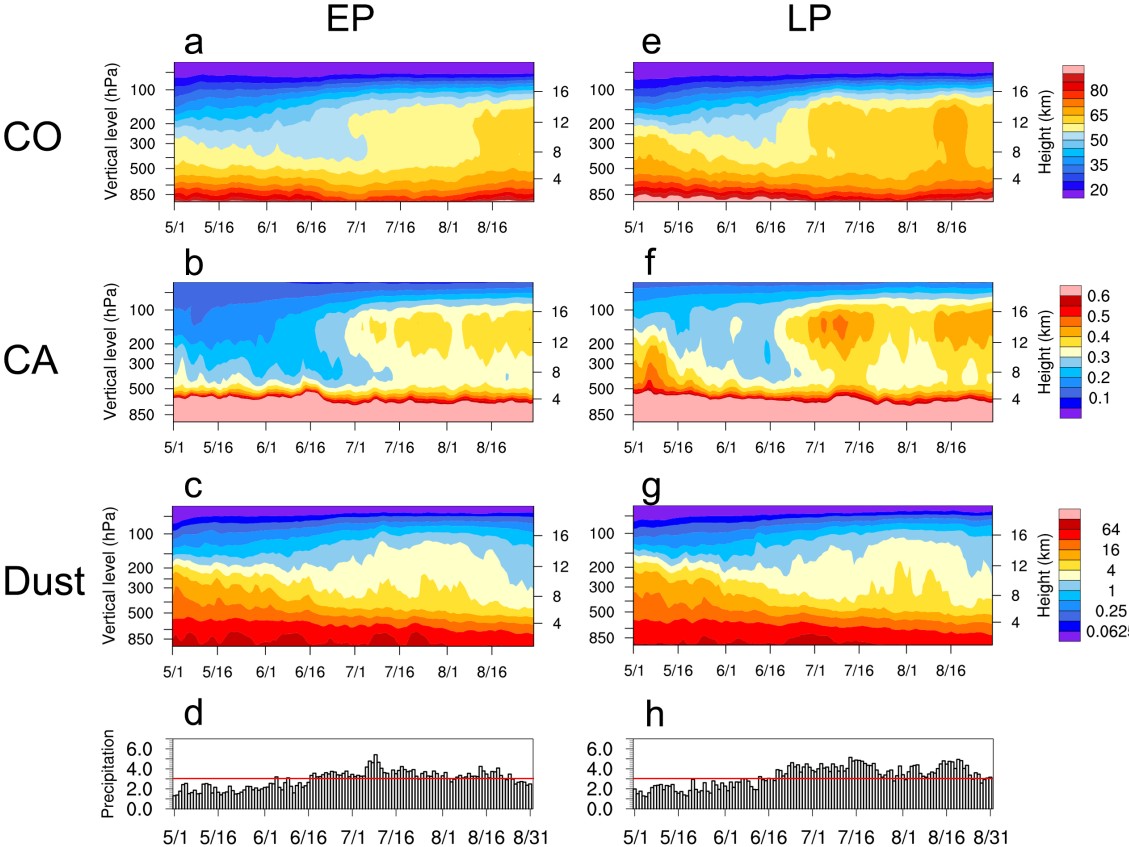

**Figure 9.** Time-height cross-sections showing daily variations in (a) CO (ppbv), (b) CA (ppbm), (c) dust (ppbm), and (d) precipitation (mm day$^{-1}$) during Late Part years over the DSCC stem regions (25°N-35°N, 65°E-115°E). Panels (e), (f), (g), and (h) are the same as (a), (b), (c), and (d) but for Early Part years. Red lines in (d) and (h) show the reference value of precipitation intensity (3 mm day$^{-1}$).