# Peer review of "Relationship between Asian monsoon strength and transport of surface aerosols to the Asian Tropopause Aerosol Layer (ATAL): Interannual variability and decadal changes"

_Atmospheric Chemistry and Physics, 2018_

## Referee Comment (RC1) · Anonymous Referee #3 · 16 Oct 2018

This study utilizes the MERRA2 reanalysis data to explore the influence of the Asian monsoon system on the Asian aerosol layer near UTLS. The topic is important, as aerosols in such high altitude likely get involved in the long-range transports and exert radiative impacts over the other regions around the world. It is also the first time to exploit MERRA2 in such a topic, even though the uncertainty of MERRA2 aerosol product remains to be gauged. The main finding about stronger Asian monsoon resulting in more abundant aerosols near UTLS reveals the relative importance of two competing mechanisms, i.e. the enlarged convective transport and enhanced precipitation washout in those stronger monsoon year. Therefore, I recommend accepting the manuscript by ACP pending minor revisions.

1) One caveat of using MERRA2 aerosol product is its simplistic treatment of aerosol mixing state in its aerosol model GOCART, as GOCART assumes all aerosol types are externally mixed. Recent modeling studies [e.g. Wang et al., 2018, JAMES] have suggested that the mixing state and aerosol aging processes in GCM or CTM can largely change the aerosol lifetime, and consequently affect the amount of aerosols lifted to UTLS. Recent aerosol optical measurements further supported that even mineral dust can be coated by a significant amount of anthropogenic aerosols over East Asia [Tian et al., 2018, ACP]. Therefore, such a caveat in data and possible implications should be discussed in the paper.

2) Fig. 2a, the differences between red and blue contour lines are not clear. Can you find a better way to present them? Simply plot the differences of Z100? Fig. 2b is too small to see the details. Please consider to enlarge it.

3) For the interannual variability. I think the monsoon strength is definitely linked with some other climate natural variability, such as ENSO. It would be interesting to see some correlation analyses between the AMA strength, aerosol loading, and some natural variability indices, and trend analyses after those natural variabilities get removed.

4) L114-115, it should be pointed out that this sentence cannot be applied to Fig. 1 as the anomaly definition there is different with the other plots.

5) Some typos: L105, annual mean L159, over the western sector

---

## Referee Comment (RC2) · Anonymous Referee #4 · 11 Nov 2018

This study uses the 15-year (2001-2015) NASA MERRA2 reanalysis data to investigate the interannual variability and the decadal trend of CO, carbonaceous aerosol, and dust in the Asian tropopause aerosol layer and their relationship to the Asian summer monsoon strengths during the 15-year time period. While this topic is interesting, I have some major concerns of the methods that lead to the conclusions (see below). I recommend authors reexamine the methods and revise the manuscript accordingly.

1. MERRA2 aerosol of individual species is not "reanalysis data". For aerosols, only column AOD from MODIS and MISR have been assimilated in MERRA2, so it is appropriate to call the MERRA2 AOD as a reanalysis product. However, concentrations and AOD from individual aerosol species, such as CA and dust used in this paper, are not a part of reanalysis (more on that in comment #2 below). CO is completely from the model simulation without any assimilation of any observations. This aspect should be clearly stated that the datasets used in this study are not "reanalysis" datasets.

2. MERRA2 aerosol species concentrations are not appropriate for intenrannual variability and long-term trend analysis. The reason is that the MERRA2 system had to adjust the model simulated total AOD to be close to the satellite observations during assimilation, but there is no speciated aerosol information from satellite data to allow changes of aerosol composition. As a result, all model simulated aerosol species had to be adjusted by the same factor, which can introduce artifacts for increase or decrease of individual aerosol mass or AOD. Such artifacts have been clearly demonstrated in Randles et al., 2017 (Fig. 5 for example). Therefore, the interannual variability or long-term trends of individual aerosol species inferred from MERRA2 might be contaminated by the introduction of the non-physical corrections of individual aerosol species amount to match the total AOD from satellite during the assimilation process. One important practice is to take a look at the so-called "increments" from MERRA2 to see the interannual variability and trends of these increments for individual aerosol species and to assess what impacts the increments might have on the apparent dust and CA interannual variability and long-term trends.

3. Definition of strong and weak monsoon years does not seem to be appropriate. This study uses the total precipitation amount within a selected region as a measure of monsoon strength, which is certainly one of the commonly used methods to define the monsoon strength. What does not seem to be appropriate is that the strength of the ASM is not based on the total precipitation amount but is based on the detrend anomaly of precipitation amount. For example, according to Fig 1c, 2015 is a weak monsoon year with a strength weaker than 2002. However, from Fig.1b, the precipitation in 2015 is 0.5 mm/day above the 2001-2015 average while that in 2002 is about 1.9 mm/day

below the 15-year average, meaning that the JJA precipitation in 2015 is about 2.4 mm/day more than that in 2002, thus a much stronger monsoon year. If the total precip amount is the criteria for indicating the SM strength, then the determination of strong or weak monsoon years should stick with that definition, not the detrend anomaly.

4. There is a lack of evaluation of the MERRA2 products used in this study to assess the quality of these products. Although observations of dust and CA in the ATAL region is rather limited (there are some aircraft data, though), MLS on Aura satellite has been producing CO in the UTLS since 2004. I wonder if the authors can take a look at the MLS data to see if they are showing similar interannual variability and decadal trend?

Given the above concerns, I think a substantial revision is needed. While MERRA2 is a valuable reanalysis dataset for total AOD, it is not particularly suitable for quantify the interannual variability and trends of individual aerosol species for the reasons I stated above. At least the non-physical aerosol mass change issues associated with AOD assimilation should be addressed/examined since it is particularly relevant to the interannual variability of speciated aerosol concentrations. Meanwhile, the definition of "strong" and "weak" monsoon years should be reconsidered.

---

## Author Comment (AC1) · 11 Nov 2018

This study utilizes the MERRA2 reanalysis data to explore the influence of the Asian monsoon system on the Asian aerosol layer near UTLS. The topic is important, as aerosols in such high altitude likely get involved in the long-range transports and exert radiative impacts over the other regions around the world. It is also the first time to exploit MERRA2 in such a topic, even though the uncertainty of MERRA2 aerosol

product remains to be gauged. The main finding about stronger Asian monsoon resulting in more abundant aerosols near UTLS reveals the relative importance of two competing mechanisms, i.e. the enlarged convective transport and enhanced precipitation washout in those stronger monsoon years. Therefore, I recommend accepting the manuscript by ACP pending minor revisions.

1) One caveat of using MERRA2 aerosol product is its simplistic treatment of aerosol mixing state in its aerosol model GOCART, as GOCART assumes all aerosol types are externally mixed. Recent modeling studies [e.g. Wang et al., 2018, JAMES] have suggested that the mixing state and aerosol aging processes in GCM or CTM can largely change the aerosol lifetime, and consequently affect the amount of aerosols lifted to UTLS. Recent aerosol optical measurements further supported that even mineral dust can be coated by a significant amount of anthropogenic aerosols over East Asia [Tian et al., 2018, ACP]. Therefore, such a caveat in data and possible implications should be discussed in the paper.

Response: We are grateful for your helpful suggestions. We have carefully read these papers provided by you and add some words and these two references acknowledging the related issues dear to the aerosol communities in the last paragraph. "Moreover, recent modeling studies have suggested that the mixing state and aging processes can largely change the aerosol lifetime during simulation, and consequently affect the amount of aerosols lifted to UTLS, and some optical measurements further support that dust aerosol can be coated by anthropogenic aerosols over East Asia and then significantly enhance absorbing ability (Wang et al., 2018, Tian et al., 2018)."

2) Fig. 2a, the differences between red and blue contour lines are not clear. Can you find a better way to present them? Simply plot the differences of Z100? Fig. 2b is too small to see the details. Please consider to enlarge it.

Response: Figure 2 is plotted to simply present the AMA in strong monsoon years are stronger and northward shifted than that in weak monsoon years. Differences of

geopotential height at Z100 has been plotted in Figure 4b, 4d, and 4f. For Figure 2b, we have changed the composition to make it larger. The updated Figure 2 has been attached.

3) For the interannual variability. I think the monsoon strength is definitely linked with some other climate natural variability, such as ENSO. It would be interesting to see some correlation analyses between the AMA strength, aerosol loading, and some natural variability indices, and trend analyses after those natural variabilities get removed.

Response: The strength of monsoon should be linked with some other climate natural variabilities, especially in the long-term scale. ENSO, being a major source of IAV for the monsoon, will certainly contribute to the strength and weakness of the monsoon, as well as aerosol emission, loading, distribution (through winds and precipitation) in monsoon regions. However, this is not the focus of this paper. This paper is focused on the relationship between monsoon strengths, the transport of aerosols from the surface to the ATAL, on interannual to decadal times scales, thus the objective of this paper is not on "what causes the IAV of the monsoon", but rather on "how monsoon IAV can affect aerosol loading and transport to the ATAL". There are others who are already doing research along this line, such as,

Kim, M. K., Lau, W. K. M., Kim, K. M., Sang, J., Kim, Y. H., and Lee, W. S.: Amplification of ENSO effects on Indian summer monsoon by absorbing aerosols, Climate Dynamics, 46, 2657-2671, 10.1007/s00382-015-2722-y, 2016.

Abish, B., and Mohanakumar, K.: Absorbing aerosol variability over the Indian subcontinent and its increasing dependence on ENSO, Global and Planetary Change, 106, 13-19, https://doi.org/10.1016/j.gloplacha.2013.02.007, 2013.

We use only 15 years data here and it may not be sufficient for studying the long-term variabilities (normally they use decades of years data). Also, with such limited data, it is not easy to separate the effect from short- or long-term natural variability. In the following research, we will expand our data and tried to do a separated research

focusing on the long-term monsoon (such as ENSO) variabilities affecting the ATAL formation.

4) L114-115, it should be pointed out that this sentence cannot be applied to Fig. 1 as the anomaly definition there is different with the other plots.

Response: We have added some words in this sentence in order to avoid misleading. The sentence has been changed to 'Henceforth, the term "anomaly" in the following parts refers to the difference between SM and WM composites (SM minus WM).'

5) Some typos: L105, annual mean L159, over the western sector

Response: All of the typos have been corrected.

[Figure]

[Figure]

**Fig. 1.** Updated Figure 2

---

## Author Comment (AC2) · 30 Nov 2018

This study uses the 15-year (2001-2015) NASA MERRA2 reanalysis data to investigate the interannual variability and the decadal trend of CO, carbonaceous aerosol, and dust in the Asian tropopause aerosol layer and their relationship to the Asian summer monsoon strengths during the 15-year time period. While this topic is interesting, I

have some major concerns of the methods that lead to the conclusions (see below). I recommend authors reexamine the methods and revise the manuscript accordingly.

1) MERRA2 aerosol of individual species is not "reanalysis data". For aerosols, only column AOD from MODIS and MISR have been assimilated in MERRA2, so it is appropriate to call the MERRA2 AOD as a reanalysis product. However, concentrations and AOD from individual aerosol species, such as CA and dust used in this paper, are not a part of reanalysis (more on that in comment #2 below). CO is completely from the model simulation without any assimilation of any observations. This aspect should be clearly stated that the datasets used in this study are not "reanalysis" datasets.

Response: We are grateful for your helpful comment and suggestions. We know that it is the total aerosol loading from the satellite that was assimilated data in the MERRA2, while their precursors and components were simulated by the GOCART model. Being the most widely used global chemical model, we'd regard the model's outcome in terms of the breakdown of the proportions of aerosol species being most plausible, or to the best of knowledge available at least, as the model results have been assessed extensively. Of course, any model results can only be trusted to the extent to which actual measurements are input to the model. For the data of dust and carbonaceous aerosols, observational data are too limited (impossible to find observational data with the same kind of spatial and temporal resolution as MERRA2) to be used for our analysis. We have stated the limitations in the revised manuscript (see below).

2) MERRA2 aerosol species concentrations are not appropriate for intenrannual variability and long-term trend analysis. The reason is that the MERRA2 system had to adjust the model simulated total AOD to be close to the satellite observations during assimilation, but there is no speciated aerosol information from satellite data to allow changes of aerosol composition. As a result, all model simulated aerosol species had to be adjusted by the same factor, which can introduce artifacts for increase or decrease of individual aerosol mass or AOD. Such artifacts have been clearly demonstrated in Randles et al., 2017 (Fig. 5 for example). Therefore, the interannual variability or long-term trends of individual aerosol species inferred from MERRA2 might be contaminated by the introduction of the non-physical corrections of individual aerosol species amount to match the total AOD from satellite during the assimilation process. One important practice is to take a look at the so-called "increments" from MERRA2 to see the interannual variability and trends of these increments for individual aerosol species and to assess what impacts the increments might have on the apparent dust and CA interannual variability and long-term trends.

Response: We agree with the limitation as stated which is clearly acknowledged in the revised manuscript. It is worth noting that we did not address the issues of climate change, but rather the variability (IAV and IDV/trend) driven by emissions, dynamical, physical and chemical processes, all of which are subject to changes spatially and temporally. Considerable efforts have made to assure their general soundness as many influential variables have been constrained by observations. In our previous research (Lau et al., 2018), we have validated the AOD, aerosol vertical distributions and precipitation from MERRA2 with MODIS, CALIPSO, and GPCP, respectively. Moreover, we have compared the CO horizontal distribution in the UTLS with MLS observation, results of comparison look good as well. Per your suggestion, we have conducted incremental analysis and found that the corrections are generally non-physical with little impact on our major conclusion. The following paragraph is added in the part of Summary:

There are limitations in using the MERRA2 aerosol species concentrations for intenrannual variability and long-term trend analysis. The MERRA2 system adjusts the model simulation according to the total AOD retrieved from satellite measurements during assimilation, but there is no speciated aerosol information from satellite data to allow changes of aerosol composition which is simulated by the widely-used chemical model of GOGART (Chin, 2000, 2002, 2016; Kim, 2017). As a result, all model simulated aerosol species had to be adjusted by the same factor, which can introduce artifacts for an increase or decrease of individual aerosol mass or AOD (Randles et al., 2017).

[Figure]

To test if the interannual variability or long-term trends of individual aerosol species inferred from MERRA2 might be contaminated by any non-physical corrections of individual aerosol species during the assimilation process. We have taken a look at the 'increment' for CA (BC+OC) and DU (Dust) from the MERRA2 dataset. Results show that in our research domain, the assimilation increments for CA and Dust aerosols are very small. In most cases, it is nearly zero and the ratio of the rest increment to the values of the model mean signal is less than 1%. Therefore, the model aerosol physics are likely to be reasonable.

Results shown in this paper are the beginning, and not final, which are useful to provide a better understanding in the context of model monsoon physics and aerosol processes and in providing guidance for future data analysis. When better and more data are available, our approach would be valuable for any follow-on pursuit on the same issue.

For the precipitation, MERRA2 provides model simulated and observations-based products, and it has been assimilated and validated with both GPCP and TRMM data, more details can be found in Reichle et al., (2017). For this research, we have validated our calculation in Figure 1b and 1c with TRMM, the result for comparison has been shown below. Similar results can be found from TRMM observational data analysis.

3) Definition of strong and weak monsoon years does not seem to be appropriate. This study uses the total precipitation amount within a selected region as a measure of monsoon strength, which is certainly one of the commonly used methods to define the monsoon strength. What does not seem to be appropriate is that the strength of the ASM is not based on the total precipitation amount but is based on the detrend anomaly of precipitation amount. For example, according to Fig 1c, 2015 is a weak monsoon year with a strength weaker than 2002. However, from Fig.1b, the precipitation in 2015 is 0.5 mm/day above the 2001-2015 average while that in 2002 is about 1.9 mm/day below the 15-year average, meaning that the JJA precipitation in 2015 is about 2.4 mm/day more than that in 2002, thus a much stronger monsoon year. If the total precip

amount is the criteria for indicating the SM strength, then the determination of strong or weak monsoon years should stick with that definition, not the detrend anomaly.

Response: Using the intensity of precipitation within a select region for separating IAV, IDV/trend is a very common method. In our analysis, the data record is relatively short when compared to other IDV climatological research, thus we cannot separate IDV and long-term trend. Our trend could be a part of the longer IDV, and probably contains emission (anthropogenic) effects, which may be reflected in the increasing monsoon strength itself, but we cannot isolate the emission effect directly since the emission inputs are not updated properly. In our analysis, the IAV variability is based on the detrended dataset and the trend based on the last 7 years compared with the first 7 years. An analogous approach is to identify the most dominant modes and the trend using EOF analysis. However, because of the short length of the dataset, the trend signal usually does not come out as a single mode, but always mixed with IAV and IDV. We used the composite and linear trend approach because it is simpler and more intuitive. We are careful in the paper, not to attribute causes to the trend, but rather say they are consistent with the IAV of monsoon strength as similarly defined, but based on separation time scales. The important point is that the strong years selected from the first (most dominant mode) may not be aligned exactly with those selected from the raw data. Because if we focus on the IAV, we don't want it to be contaminated by the "trend", at least in the linear sense, and vice versa. There is plenty of recent and past paper, where IAV, IDV and trend signals of the monsoons are separated by EOFs and/or methods similar to ours. The following is a few examples:

1. Chang C P, Zhang Y, Li T. Interannual and interdecadal variations of the East Asian summer monsoon and tropical Pacific SSTs. Part I: Roles of the subtropical ridge[J]. Journal of Climate, 2000, 13(24): 4310-4325.

2. Singhrattna N, Rajagopalan B, Kumar K K, et al. Interannual and interdecadal variability of Thailand summer monsoon season[J]. Journal of Climate, 2005, 18(11): 1697-1708.

3. Wang B, Wu Z, Chang C P, et al. Another look at interannual-to-interdecadal variations of the East Asian winter monsoon: The northern and southern temperature modes[J]. Journal of Climate, 2010, 23(6): 1495-1512.

4. Giannini A, Saravanan R, Chang P. Oceanic forcing of Sahel rainfall on interannual to interdecadal time scales[J]. Science, 2003, 302(5647): 1027-1030.

4) There is a lack of evaluation of the MERRA2 products used in this study to assess the quality of these products. Although observations of dust and CA in the ATAL region is rather limited (there are some aircraft data, though), MLS on Aura satellite has been producing CO in the UTLS since 2004. I wonder if the authors can take a look at the MLS data to see if they are showing similar interannual variability and decadal trend?

Response: We have done some validation based on observational data in our previous research (Lau et al., 2018). Per your suggestions, we have used the MLS CO data to verify our results, and the zonal cross-section of anomaly between SM (2007, 2010, 2011, 2013) and WM (2014, 2015) years is shown below. The concentration of CO is increased from mid-troposphere to the UTLS region, implying the transportation is enhanced during SM years.

Figure

Figure Caption. Longitude-height cross-sections (0°E-140°E) of CO (ppbv) anomaly between strong and weak monsoon years ('strong' minus 'weak') averaged over the southern portion of the AMA (25°N-35°N) during July-August.

It is worthy to be noted that the MLS CO data record is too short (only provide data after 2004 August), thus we don't have enough samples of SM and WM years, or EP and LP years as defined to do a meaningful IAV composite, and trend analysis. Also, it has been suggested by other research that the MLS CO has up to 30% uncertainties at 100 hPa (Livesey, 2008; Santee, 2017), thus in this case, single year data with large anomalies may dominate the mean. What's more, if there is a large change in

anthropogenic emissions, those changes will not be captured by MERRA2, because the emission inventories are not updated since the mid-2000s in the model. Therefore, the availability of observational data for certain type of aerosol is highly expected from us to use in further research to validate our current results.

Livesey, N. J., et al. (2008), Validation of Aura Microwave Limb Sounder O3 and CO observations in the upper troposphere and lower stratosphere, J. Geophys. Res., 113, D15S02, doi: 10.1029/2007JD008805.

Santee, M. L., G. L. Manney, N. J. Livesey, M. J. Schwartz, J. L. Neu, and W. G. Read (2017), A comprehensive overview of the climatological composition of the Asian summer monsoon anticyclone based on 10 years of Aura Microwave Limb Sounder measurements, J. Geophys. Res. Atmos., 122, 5491–5514, doi: 10.1002/2016JD026408.

[Figure]

**With trend**

**After detrend**

**MERRA 2**

**TRMM**

**Fig. 1.** Comparison of trend of precipitation from MERRA2 and TRMM

[Figure]

**Fig. 2.** Zonal cross-section of MLS_CO between 25 N and 35 N

---

## Referee Report (RR1)

Review of revised manuscript, "Relationship between Asian monsoon strength and transport of surface aerosols to the Asian Tropopause Aerosol Layer (ATAL): Interannual variability and decadal changes" by Cheng Yuan et al.

I appreciate that the authors have addressed all my original comments and revised the manuscript. Out of the four comments, two of them still need to be considered or better explained:

a) Definition of "strong" and "weak" monsoon years: I think the authors may have misunderstood my concerns. I have no problem to use the precipitation amount as an indicator for Asian summer monsoon strength. In that regard, the precipitation amount, or the anomaly of the precipitation amount, should be used as a measure for the monsoon strength, rather than the "detrend" anomaly. For example, why is 2015 a weaker monsoon year than 2002 even though the precipitation amount in 2015 is about 2.3 mm/day more than that in 2002?

b) Model evaluation with data: In the response (page 7, line 233-239), the authors said that the MLS CO data record is too short to have enough samples of SM and WM years. But my point was not to use the MLS CO data for SM and WM contrast, but for comparisons with the model interannual variabilities and UTLS CO magnitude. Besides, the MLS CO data record (11.5 years) is only 3.5 years shorter than the MERRA2 results (15 years) used in this study. In addition to CO, the precipitation from MERRA2 (which is not a reanalysis either) can also be compared with satellite data such as TRMM (shown in the response). Clearly, MERRA2 precipitation trend are much larger than the TRMM trend by almost a factor of 5! These comparisons should be shown in the paper (or supplement if appropriate) to let the readers have some ideas about the MERRA2 products.

---

## Author Response (AR2)

**Dear Editor,**

**We are very grateful for the reviewer #4's comments. The responses to comments are provided below, and the manuscript has been revised accordingly. The reviewers' comments and suggestions are highly valuable in improving the quality of our manuscript greatly.**

**We are looking forward to your response.**

**Yours sincerely,**

**Cheng Yuan[1,2], William K. M. Lau[2,3], Zhanqing Li[2,3], Maureen Cribb[2], Tijian Wang[1]**

**[1] School of Atmospheric Sciences, Nanjing University, Nanjing, 210023, China**
**[2] Earth System Science Interdisciplinary Center, University of Maryland, College Park, MD, 20740, USA**
**[3] Department of Atmospheric and Oceanic Sciences, University of Maryland, College Park, MD, 20740, USA**

**Report**

Review of revised manuscript, "Relationship between Asian monsoon strength and transport of surface aerosols to the Asian Tropopause Aerosol Layer (ATAL): Interannual variability and decadal changes" by Cheng Yuan et al.

I appreciate that the authors have addressed all my original comments and revised the manuscript. Out of the four comments, two of them still need to be considered or better explained:

**1. Definition of "strong" and "weak" monsoon years: I think the authors may have misunderstood my concerns. I have no problem to use the precipitation amount as an indicator for Asian summer monsoon strength. In that regard, the precipitation amount, or the anomaly of the precipitation amount, should be used as a measure for the monsoon strength, rather than the "detrend" anomaly. For example, why is 2015 a weaker monsoon year than 2002 even though the precipitation amount in 2015 is about 2.3 mm/day more than that in 2002?**

*Response:* As we have explained previously, over a given period, the strengths of the monsoon could be due to intrinsic variability, as reflected in IAV, as well as anthropogenic forcing, as reflected by the linear trend. The focus of the present work, similar to many previous studies, is to determine if the fundamental processes e.g. dust emission caused by monsoon wind changes over the desert, transport, dry and wet deposition of desert dust, BC and OC contributing to the intrinsic variability i.e., IAV of the monsoon may also contribute to the trend signal. To isolate the IAV, the strength of the monsoon needs to be defined using the detrended data. The reason that 2015 is considered a weak intrinsic monsoon year, even though it has more total rainfall than 2002, is because of the contribution from the trend signal, which is quite strongly positive throughout the period. These two years happens to near the beginning and the end of the record. Our results, show that changes in precipitation and transport processes contributing to the IAV of the monsoon, may also contribute to the trend signal. Of course, such an approach has obvious limitations. First the IAV and trend signals are not linear, and not simply additive. Second, the data record we used is too short, so the "trend" signal could be a part of an intrinsic signal related to natural intedecadal variability (IDV) of the system. Additionally, the emission inventory used in MERRA2 was not updated after mid-2000's. Therefore, while we can conclude that changes in monsoon transport processes contribute to both IAV and trend signal in the data record, we cannot draw any conclusions regarding roles of anthropgenic emission on the trend.

**2. Model evaluation with data: In the response (page 7, line 233-239), the authors said that the MLS CO data record is too short to have enough samples of SM and WM years. But my point was not to use the MLS CO data for SM and WM contrast, but for comparisons with the model interannual variabilities and UTLS CO magnitude. Besides, the MLS CO data record (11.5 years) is only 3.5 years shorter than the MERRA2 results**

**(15 years) used in this study. In addition to CO, the precipitation from MERRA2 (which**
**is not a reanalysis either) can also be compared with satellite data such as TRMM**
**(shown in the response). Clearly, MERRA2 precipitation trend are much larger than the**
**TRMM trend by almost a factor of 5! These comparisons should be shown in the paper**
**(or supplement if appropriate) to let the readers have some ideas about the MERRA2**
**products.**

*Response:* For the results of CO, we have compared the 2005-2015 CO concentration at 100
hPa from MLS and MERRA2 following your suggestions, and the figure for comparison has
shown below. The difference can attribute to the uncertainty from MLS observation
(mentioned in the previous response), as well as the bias from emission used in the MERRA2
simulation process. This figure has been added in the SI as Figure S4, and one sentence of
'Similar trend of CO is also seen in the results from MLS observation, and the difference in
certain year can attribute to bias from observation and the emission inventories used in
simulation (Figure S4).' has been added at L313.

[Figure]

For the results of precipitation, the rainfall of selected region from MERRA2 has been
compared with TRMM observational data. Apart from the anomalies which have been shown
in the previous response, the actual intensities of precipitation from both MERRA2 and
TRMM over the selected region has been added (see below). The precipitation from
MERRA2 is around 10-20% under estimated when compared with observational data. The
purpose of showing anomalies is to validate the increasing trend during recent years, though
the rate of increasing is different between TRMM and MERRA2. The sentence at L207 has
been changed to 'This trend has been validated by observational data (Figure S1), and a
simialr increasing decadal trend of the SASM has been reported in previous studies (Jin and
Wang, 2017)'.

[Figure]

**List of changes in the manuscript**

| Line number | Original | Changed |
|---|---|---|
| 206 | A simialr increasing decadal trend of the SASM has been reported in previous studies (Jin and Wang, 2017). | This trend has been validated by observational data (Figure S1), and a simialr increasing decadal trend of the SASM has been reported in previous studies (Jin and Wang, 2017). |
| 235 | S1 | S2 |
| 254 | S1 | S2 |
| 274 | S1b | S2b |
| 276 | S1d | S2d |
| 288 | S2 | S3 |
| 296 | S1b | S2b |
| 299 | S1d | S2d |
| 304 | S2 | S3 |

[revised manuscript text omitted]